# How Income and Income Inequality Drive Depressive Symptoms in U.S. Adults, Does Sex Matter: 2005–2016

**DOI:** 10.3390/ijerph19106227

**Published:** 2022-05-20

**Authors:** Hossein Zare, Nicholas S. Meyerson, Chineze Adania Nwankwo, Roland J. Thorpe

**Affiliations:** 1Department of Health Policy and Management, Johns Hopkins Bloomberg School of Public Health, Baltimore, MD 21205, USA; nmeyers3@jh.edu (N.S.M.); cnwankw6@jh.edu (C.A.N.); 2School of Business, University of Maryland Global Campus (UMGC), Adelphi, MD 20774, USA; 3Department of Health, Behavior, and Society, Johns Hopkins Bloomberg School of Public Health, Baltimore, MD 21205, USA; rthorpe@jhu.edu

**Keywords:** income, income inequality, depression, poverty to income ratio, PHQ-9

## Abstract

Importance: Depression is one of the leading causes of disability in the United States. Depression prevalence varies by income and sex, but more evidence is needed on the role income inequality may play in these associations. Objective: To examine the association between the Poverty to Income Ratio (PIR)—as a proxy for income—and depressive symptoms in adults ages 20 years and older, and to test how depression was concentrated among PIR. Design: Using the 2005–2016 National Health and Nutrition Examination Survey (NHANES), we employed Negative Binomial Regression (NBRG) in a sample of 24,166 adults. We used a 9-item PHQ (Public Health Questionnaire, PHQ-9) to measure the presence of depressive symptoms as an outcome variable. Additionally, we plotted a concentration curve to explain how depression is distributed among PIR. Results: In comparison with high-income, the low-income population in the study suffered more from greater than or equal to ten on the PHQ-9 by 4.5 and 3.5 times, respectively. The results of NBRG have shown that people with low-PIR (IRR: 1.30, 95% CI: 1.23–1.37) and medium-PIR (IRR: 1.55, 95% CI: 1.46–1.65) have experienced a higher relative risk ratio of having depressive symptoms. Women have a higher IRR (IRR: 1.29, 95% CI: 1.24–1.34) than men. We observed that depression was concentrated among low-PIR men and women, with a higher concentration among women. Conclusion and Relevance: Addressing depression should target low-income populations and populations with higher income inequality.

## 1. Introduction

Depression is one of the most common mental health disorders in the United States, with 18.5% of adults (ages 18 years and older) reporting symptoms in 2019 [1]; 15% of men and 21.8% of women have reported depressive symptoms [1] with a one percentage point increase between 2005 and 2016 [2]. In 2019, nearly 20 million U.S. adults experienced at least one major depressive episode [3]. The prevalence of a major depressive episode was nearly 1.5 times greater among women (9.6%) compared to men (6.0%) [3]. Research has attributed these gender differences to females experiencing more life stressors, the role of puberty, and gender inequalities [4,5,6,7].

Changes in environmental and socio-economic factors are commonly associated with an increased risk of depression [8,9,10]. The prior literature has identified education, income status, BMI, smoking, and health conditions (i.e., hypertension and diabetes) as risk factors for the increase in the incidence of depression [11,12,13,14,15] and depression severity [4].

Researchers, especially economists and social epidemiologists, have long been interested in the relationship between income and mood [16,17,18]. Despite the enduring interest in this area, few studies have examined the association between income and depression [17,19,20,21]. Prior literature has highlighted the prevalence of depression among low-income individuals [17,22]. For example, a study among women receiving welfare assistance found one out of five met the criteria for Major Depressive Disorder (MDD) [23]. Moreover, low-income individuals are at increased odds of depression compared to individuals who report higher income [17]. Between 2013 and 2016, approximately 16% of adults living below the federal poverty line had depression, compared to 3.5% of adults at or above 400% of the federal poverty line [24].

Several mechanisms have been suggested to explain the relationship between income and depression. Low-income individuals experience more adverse effects from their neighborhood. They also experience a higher concentration of crime, less access to medical/psychological services, and fewer healthier food options; and increased blight could worsen mental health [25,26,27,28,29]. According to stress theory, financial and housing insecurities can increase the risk of mental health disorders [30,31]. Financial loss from retirement can lead to lower incomes for elderly populations and the loss of restriction of social interactions might worsen depressive symptoms [32]. Having a lower-income status has been commonly associated with worse physical health [33,34,35]. Robust studies have highlighted that chronic conditions are associated with depressive disorders [36,37].

In recent years, researchers have become increasingly interested in how the distribution of income across society affects depression. One theoretical perspective postulated that as income disparities increase, low-income individuals experience financial insecurity and shame [38,39]. These negative emotions, associated with their economic standing, lead to self-blame, stress, and social isolation, thus increasing their risk of developing depression [40]. Moreover, income inequality promotes forms of material deprivation. A lack of investment in low-income areas (i.e., housing, education, and public transportation) exacerbate other inequities that lead to worse physical and mental health outcomes [40,41]. A recent meta-analysis found the risk of depression was 1.19 times greater in populations with higher income inequality compared to populations with lower income inequality [40].

A few studies have examined the association between income and depression; however, fewer studies have stratified analyses by sex. Among these studies, only three were based in the U.S., of which one was conducted only among adolescents [42] and the other two used less than 3 years of data [43,44]. The present study is the first to examine the relationship between income inequality and depression and sex in a wide range of NHANES data (2005–2016). We used weighted models that made our findings nationally representative estimates, increasing the generalizability of these results.

This study will look at the association between the Poverty to Income Ratio and depressive symptoms and how this relationship may operate differently in men and women. In addition, using the concentration curve, we will estimate the concentration of depression among PIR in men and women to plot how the distribution of income across society affects depression. The results of this study will inform policymakers about the need to address income differences, the distribution of income across society, and their impact on depressive symptoms.

## 2. Methods

### 2.1. Data and Study Population

Data for this study came from the 2005–2016 National Health and Nutrition Examination Survey (NHANES) [45], a cross-sectional survey that provides nationally representative estimates of the health and nutritional status of the U.S. population. It had a response rate of 73.2% between 1999 and 2016 [46,47] and a multistage probability sampling design that makes up the sample representative of each of the four regions of the U.S. For this study, we included participants who were 20 years and older, who identified themselves as White Non-Hispanic (NH), Black NH, and Hispanic. We excluded missing observations for the Poverty Income Ratio (PIR) (1443 men and 1535 women) and PHQ-9 questions (if none of the questions had been responded to, from 1756 men and 2165 women), which yielded an analytic sample of 24,166.

### 2.2. Measures

*Dependent variable.* The nine-item version of the Public Health Questionnaire (PHQ-9) was used by NHANES to measure depressive symptoms. The Patient Health Questionnaire-9 (PHQ-9) measurement tool assesses the severity of depression on a graded scale and is a brief version of the Primary Care Evaluation of Mental Disorders (PRIME-MD). The self-administered questionnaire by the patient is a reliable and valid measurement and states that a score > 9 has an 88% sensitivity and specificity, making it best for clinical use. Scores of 5, 10, 15, or 20 represent mild, moderate, moderately severe, and severe depression, respectively, on the DSM-IV scale [11]. The PHQ-9 is a clinically validated survey with a sensitivity and specificity of 88% [8]. Respondents rated how often over the past two weeks they experienced depressive symptoms, such as restless sleep, poor appetite, and feeling lonely. Each item was scored on a 4-point ordinal scale for frequency (0, not at all; 1, several days; 2, more than half the days; 3, nearly every day). The total score was calculated by finding the sum of nine items; this approach yielded a maximum score of 27 [48]. We used this scale as the dependent variable in all NBRG models. Additionally, to run a logistic model, we created a dummy variable with a cutoff score of 10 or higher (=1, if PHQ ≥ 10, =0, if PHQ < 10).

### 2.3. Main Independent Variable

*Poverty Income Ratio (PIR).* We used PIR as the main independent variable as a proxy of income. The NHANES calculated the PIR by dividing family (or individual) income by the poverty guidelines specific for each survey year. Comparing to income explains more about the family income by including the family income, family size, and the poverty threshold, providing more reliable socioeconomic status of a family. For example, for a family of four with 2 children younger than 18 years old, with an annual income of USD 32,000, and a poverty threshold of USD 31,661, the PIR would be (USD 32,000/USD 31,661 = 1.01) [49]. Using this ratio, we defined a categorical variable with three categories: low-PIR (0–1.16), medium-PIR (1.17–2.82), and high-PIR (2.83–5.00) quintiles from low to high.

### 2.4. Covariate

We controlled models for demographic variables, including age (years), sex (1 = female, 0 = male), marital status (1 = married, 0 = otherwise), racial and ethnic status (White NH, Black NH, and Hispanic), and educational attainment (less than high school graduate, high school graduate or general equivalency diploma, more than high school education, or some college and above). We also controlled models for comorbidity (any type of chronic disease, including emphysema, thyroid problem, chronic bronchitis, COPD, asthma, arthritis, malignancy, stroke, diabetes, coronary heart failure, angina pectoris, heart attack, and liver problems), having health insurance (1 = yes; 0 = no), and utilization of mental health professionals. We also considered whether the interview was conducted in English (speak English well) and living alone as household characteristics.

### 2.5. Analytic Strategy

We conducted two sets of analyses. The first set of analyses focused on exploring the relationship between PIR, sex, and depression, accounting for other demographic, SES, and health-related characteristics. We used chi-squares to compare the means among PIR categories. To estimate the impact of the PIR on participants’ PHQ-9 scores, we conducted several sets of Negative Binomial Regression Models [50] to report Incidence-Rate Ratios (IRR) and the corresponding 95% confidence intervals (CI) [51,52]. To find the best fit model, first, we ran a Poisson regression model, even though we believe that the Poisson distribution is incorrect. The considerable value of goodness of fit chi-square (24103) and *p* < 0.01 indicates that the Poisson model is inappropriate [53]. After running NBRG, by looking at the likelihood ratio test (a test of the overdispersion parameter alpha), alpha was significantly different from zero (*p* < 0.01), and thus reinforces one last time that the Poisson distribution is not appropriate, and the Negative binomial regression is more appropriate in cases of overdispersion.

In the first model, we estimated the association between PIR and the participants’ PHQ-9 scale, controlling for sex, age categories, marital status, education, race, ethnicity, comorbidity, health insurance coverage, utilization of mental health professionals, speak English well, and living alone. Then, to find how sex plays a role in the association between PIR and participants’ PHQ-9 scale, we ran the second model with interactions between PIR and sex. We created a variable by interrelating sex and PIR categories, which we used in the second model. All analyses were weighted using the NHANES individual-level sampling weights for 2005–2016 (6 waves of data) [54]. As such, the estimates are representative of the national level for the U.S. civilian population [54].

For the second set of analyses, to learn how depressive symptoms concentrated among PIR in men and women, we plotted the concentrated curve of the clinically depressed population (PHQ ≥ 10, *y*-axis) against the cumulative percentage of the people across different levels of PIR. We plotted the concentration curve for men and women. We then ranked them by PIR, beginning with the lowest and ending with the wealthiest on the *x*-axis, and the 45-degree line shows perfect equality [55]. Using PIR, we plotted PIR distribution among men and women. The Lorenz curve is a well-known plot to show income distribution in each specific society. The Gini Coefficient (GC) is defined as A/(A + B): A is the area between the line of perfect equality (45-degree line) and the Lorenz curve; B is the area between the Lorenz curve *x*- and *y*-axis. If ‘A’ equals zero, then GC will be zero, which means perfect equality; if ‘B’ is zero, then the GC will be one, which means absolute inequality [56]. A combination of concentration curve and Lorenz curve show the movement of depression across the population with different levels of PIR. The Lorenz curve presents the level of inequality in a society and specific people. This analysis will help us understand how the probability of being depressed correlates with PIR. We used State version 15 for all analysis [57].

## 3. Results

### 3.1. Descriptive Analysis Results

A total number of 24,166 individuals were included in our analyses. Among all participants, 17.8% (*n* = 6,609) had low-PIR, 29.8% (*n* = 8,353) had medium-PIR and 52.3% had high-PIR (*n* = 9,204). Overall, 74.5% of participants were White NH, 11.5% Black NH, and 13.7% Hispanics. On average, participants were 47.4 years old (SD = 14.3). Among all participants, 50.6% were female, 64% were married, and 61% had a degree beyond a high school diploma (See Table 1).

Table 1, panel A shows the distribution of the PHQ-9 scale and PIR groups. As expected, with increasing PIR, the PHQ-9 scale decreased. It reduced from 16.3 in low-PIR to 9.2 in medium-PIR and 3.8 in high-PIR groups. About 35% of low-PIR have experienced some level of depression. Prevalence of depression was lower in the medium-PIR (26%) and high-PIR (16%) groups. Overall, 23% of the sample had experienced some level of depression.

Table 1, panel B compares the SES variables among PIR categories. Compared to the high-PIR group, participants in the low-PIR group were younger (43.5 ± 18), more likely to be single (55%), female (54%), educational level of less than a high school diploma (36%), and were of a racial and/or ethnic minority. Among participants in the low-PIR group, 37% were not under the coverage of any health insurance.

Compared with the high-PIR group, participants in medium-PIR groups were more likely to be female (51%) and married (73%). Compared with high-PIR, the medium-PIR group had a lower percentage of college graduates or above (15% vs. 44%), and 31% were racial and/or ethnic minorities. Among participants in the medium-PIR group, 76% had health insurance compared to 93% of the high-PIR group.

In Figure 1, we compared trends between 2005 and 2016. The low-PIR population had suffered from depression (PHQ ≥ 10) 4.9 times more than high-PIR and 1.7 times more than medium-PIR groups. Among all participants, depressive symptoms increased by 25.2% between 2005 and 2016. The prevalence of depression increased by 2.5% (from 13.8% to 16.2%) in the low-PIR population. The probability of being depressed in the medium- and high-PIR groups increased by 1.6% (6.8% to 8.4%) and 1% (2.8% to 3.8%), respectively.

### 3.2. Association between Poverty Income Ratio Level and Depressive Symptoms

The association between PIR levels and the PHQ-9 scale is displayed in Table 2, model 1. As presented, in comparison with the high-PIR group, people in low- and medium-PIR have a higher incidence-rate ratio (IRR). Per our adjusted full model, results indicate that the incident rate for low-PIR is higher than the comparison group (high-PIR) by 1.55 (CI:1.46–1.65) times. Likewise, the medium-PIR experiences a higher PHQ-9 scale by 1.30 times (CI:1.23–1.34) in comparison with the people in high-PIR, but is lower than the low-PIR groups, while holding the other variables constant. Income worked as a protective for depressive symptoms. Educational attainment, marital status, and having health insurance were protective factors against depressive symptoms.

Women suffered more from higher depressive symptoms than men (IRR: 1.29, CI: 1.24–1.34). Moreover, younger adults (34–49 years old), people with comorbidities, and people who spoke English had a higher rate of depressive symptoms. In comparison with White NH individuals, Black NH individuals had a lower rate of depressive symptoms.

### 3.3. Association between Poverty Income Ratio Levels and Depressive Symptoms in Men and Women

Table 2, model 2 presents the results of interaction between PIR and sex, using women with high-PIR as a reference group. As we see, low-PIR women and men are experiencing a higher incidence-rate ratio of 1.54 (CI:1.44–1.65) and 1.19 (CI:1.10–1.29), respectively. Income is a protective factor against depressive symptoms in men in middle-PIR groups but not in women. Finally, for the last set of analyses, and because the interaction between PIR and sex was significant (*p* < 0.001), we stratified analyses by sex (see Table 3). Our findings showed that there are some differences between SES variables in men and women. For example, education is protective only in men with college degrees, but not in women with high school degrees and above. Race and ethnicity behave differently between men and women. As shown, Black NH and Hispanic women follow the trend of White NH women, but Black NH and Hispanic men have lower incident rates by 0.86 (CI: 0.80–0.93) and 0.92 (CI: 0.85–0.99), respectively, in comparison with White NH men.

### 3.4. Depressive Symptoms and Income Inequality

Figure 2, panel A presents the concentration of depression in male populations ranked by PIR. The concentration curve for depressed is the solid blue line. As we see, the higher probability of being depressed is concentrated in the low-income population, e.g., the bottom 25% of men in the income distribution, where we observed about 34.8% of depression. It increased to 82.4% in 75% of the population. We see a similar pattern in women with a slightly higher concentration of depression in the low-income population. For example, 35.8% of depression is concentrated in the bottom 25% of women, increasing to 82.6% in 75% of the population.

The red dash line shows the distribution of PIR using the Lorenz curve. As presented by the Lorenz curve in panel A, only 7.5% of income was distributed among 25% of men, and 75% received only 59.5% of the income. We see a similar pattern and slightly higher disparities in women. For example, only 7.2% of income was distributed among 25% of women, and 75% received only 57.4% of the income.

**Sensitivity analysis.** As a sensitivity analysis, we created a dummy variable with a cutoff score of 10 or higher (=1, if PHQ ≥ 10, =0, if PHQ < 10) and run sets of logistic models. Our results showed that people in medium-PIR (OR = 0.62, CI [0.53–0.72] and high-PIR (OR = 0.36, CI [0.29–0.45] groups were less likely to have depressive symptoms, more than 10 in comparison with the low-PIR reference group. Also, women, adults 36–45 years old, and people with comorbidities suffered more from depression (See Appendix A).

## 4. Discussion

### 4.1. This Work

In this study, we investigated the relationship between depressive symptoms (using the PHQ-9 questionnaire) and income measured by the PIR in an adult U.S. population aged 20 years and older with a sample of 24,166. We also studied this association between men and women. For the first analysis set, we used NBRG regression models to show how the PHQ-scale changes between people with different levels of income and between men and women. For the second set of analyses, we plotted the concentrated curve of a clinically depressed population against the cumulative percentage of the population to show how depressive symptoms are concentrated among PIR in men and women. Several findings of this study need specific attention in addressing depression. In the following paragraphs, we discuss these findings and policy recommendations.

#### 4.1.1. Sex Differences

We observed the highest concentration of depression among the low-income population in men and women. For example, the bottom 25% of men in the income distribution have observed about 34.8% of depression, and 35.8% of depression is concentrated in 25% of low-PIR women. Based on our results, future research is needed to consider the mechanisms of these disparities experienced by women. Comparing the GC has shown that income is more unequally distributed among depressed women. Prior literature suggests women may be more susceptible to the consequences of the social isolation caused by income inequality, which exacerbates the risk of developing depressive symptoms [58,59]. Furthermore, poor access to mental health services in low-income areas may have a stronger impact on women than on men.

#### 4.1.2. Comorbidity and Depression

Often, depression is accompanied by other mental illnesses and physical disorders. More than half of individuals who have reported a lifetime history of depression have also reported a lifetime diagnosis of an anxiety disorder [60]. Furthermore, among individuals with lifetime MDD, approximately 40% had been diagnosed with an alcohol use disorder, and 17% had been diagnosed with a drug use disorder [61]. Prior studies have estimated individuals with chronic health conditions were nearly three times more likely to have MDD than individuals without chronic health conditions [61]. Low-income populations are disproportionately impacted by comorbidities [62]. Patients with comorbid disorders experience higher medical costs and greater reliance on medical services.

#### 4.1.3. Cost of Depression

There is wide variation across countries [63]. The country of residency and international differences of wages are two important predictors of estimating indirect and direct cost of disease. For example, the annual cost of MDD in Singapore is estimated at USD 7,638 but is USD 10,379 in the US. In the US, for every USD 1 of direct cost there is an additional USD 6.6 cost, including a USD 1.9 indirect cost and a USD 4.7 workplace morbidity cost [64]. Considering these types of costs highlights how a patient with MDD can go beyond the poverty line, which is crucial for low-income populations [65,66].

#### 4.1.4. Treatment

MDD is estimated to cost the US more than USD 300 billion per year [67], a significant portion of which is from lost work productivity [68]. Studies have highlighted that depressive symptoms are associated with worse presenteeism and absenteeism [68,69]. Low-income individuals are at an elevated risk of presenteeism and absenteeism [65,66]. Improving depression treatment is an effective option to improve workplace functioning. Providing treatment has been found to improve productivity and to reduce absenteeism [70,71].

#### 4.1.5. Addressing Fundamental Inequality

Depression is problematic, especially for low-income populations. We have shown in the first set of analyses that income is protective against depression. In comparison with people in upper-PIR groups, those in lower-PIR had experienced higher depressive symptoms. Additionally, women had experienced higher IRR than men; thus, any future effort to reduce depression needs to enhance its reach for women in particular, given the high need in that group. Further research in this area needs to be done focusing on the associations between income and depression with attention to the role of gender differences. Based on our findings from the literature review, little is known about the impact of income inequality and depression.

As we have shown in the second set of analyses, the income distribution is more unequal in women than in men (GC: 0.315 vs. 0.296, *p* < 0.001). It may explain some inequality such as an income gap [72,73], occupational risk factors [74], and wage disparities [75]. In developing depression-reducing policies, addressing these fundamental inequalities needs essential attention [76].

#### 4.1.6. Depression Concentration

As presented by the concentration curve, depression is concentrated in the low-income population in both men and women. It highlights the importance of providing more social assistance services to low-income communities, regardless of gender. Although lower-PIR women are experiencing higher depression, we found a similar situation in low-PIR men. The U.S. can take lessons from other countries such as Finland, the Netherlands, and New Zealand. Universal healthcare can address issues in the cost and accessibility of mental health services. Society may benefit from economic policies to improve the distribution of income. For example, universal basic income and progressive taxation are potential policies that could address the wealth gap. Policies should also consider gender and racial/ethnic inequities. Policies to mandate equal compensation for women and racial/ethnic minorities are crucial to closing the pay gap.

#### 4.1.7. The Pandemic Influence on Depression

The pandemic has increased the depression rate, targeted the global economy [77], influenced the inflation rate, and unequally impacted all countries worldwide. With the unequal distribution and utilization of vaccination, many developed countries are expected to recover much more rapidly than low-income countries [77]. All these elements may impact unweighted inequality between countries. However, we need actual data on individual incomes from household surveys to estimate that. The pandemic also has differential economic impacts on different labor force segments; people in occupations conducive to remote work were less experienced, lost their jobs and faced financial stress [78]. Moreover, many people not only experienced immediate financial pain but also anxiety and stress about the disease, among other concerns. It is fair to say that the pandemic increased income inequality, and the gap between high-wage and low-wage employees, and between men and women grew. With a positive association between the pandemic and depression, anxiety, and stress [79], and more negative impact on women [80], communities should learn mechanisms to reduce anxiety [81], and policymakers should consider mental health screenings that prioritize women and low-income populations.

### 4.2. Limitations

Several aspects of the present study deserve comment. The data were cross-sectional; therefore, we could not rule out the possibility of reverse causation. The evidence indicated that the extent of bias due to reverse causation was largely indirect [82]. The NHANES data had some limitations regarding the income variable and did not report real income; instead, income was reported as a categorical variable. Employing household income as a continuous variable could allow us to find the impact of income differences instead of a proxy variable such as the PIR. Another potential limitation was the definition of depression. Although the PHQ-9 validated the clinical depression by 82%, we have not measured clinical depression in this study. We need to note that the Lorenz curves are unaffected by the mean of the distribution, and “they cannot be used to rank distributions in terms of social welfare, only in terms of inequality [83]”.

### 4.3. Future Work

In this study, we focused on gender differences; there is a need for future studies on racial and ethnic minorities. Future research should also look at the place to find how state policies on income inequality could address depression and geographic disparities; for example, considering differences between urban and rural areas or variables such as population density, pollutants, or social vulnerability.

## 5. Conclusions

We found a strong association between depression and PIR in U.S. adults. During the last 16 years, depression has increased in all populations, with a higher increase in low-income populations. Our findings also indicate that depression is concentrated among low-PIR men and women with a higher distribution among women. Policymakers should consider a combination of local and federal policies to improve health outcomes and better distribute income. Specifically, in low-income and vulnerable communities, the resources need to not only address the depression but also provide social assistance and community-level activities to keep people with depression more involved within their community.

## Figures and Tables

**Figure 1 ijerph-19-06227-f001:**
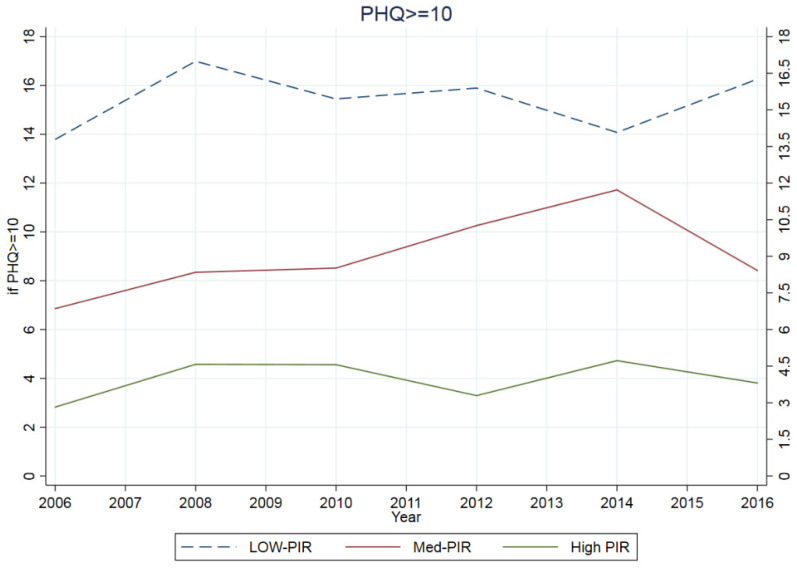
Comparing the ratio of family income to poverty in people with depression.

**Figure 2 ijerph-19-06227-f002:**
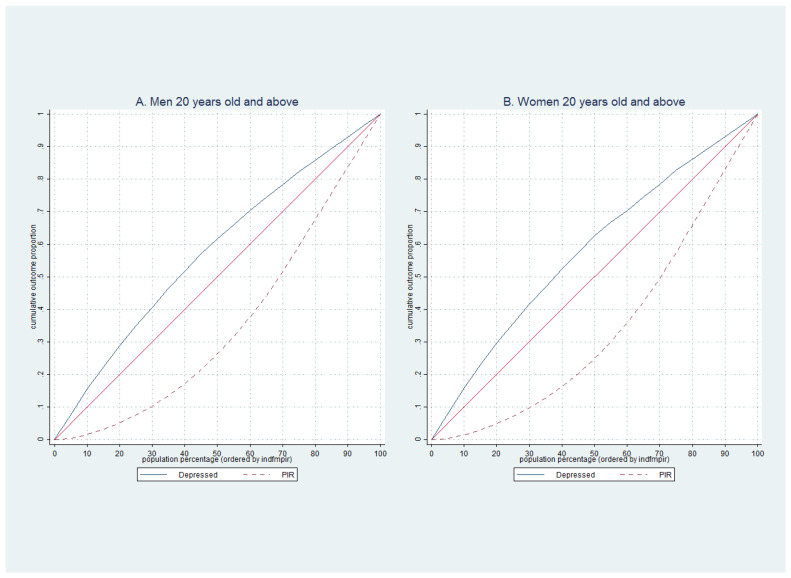
The Concentration curves to show distribution of PIR and depression in men and women between 2005 and 2016.

**Table 1 ijerph-19-06227-t001:** Demographic Characteristics of Nationally Representative Samples Aged 20 and above, NHANES 2005–2016.

	Low (N = 6525)		Medium (N *=* 8246)		High (N = 9178)		All N = 23,949	
Depressive Symptoms	Mean/%	(SD)	Mean/%	(SD)	Mean/%	(SD)	Mean/%	(SD)
**Panel A**								
PHQ-D Score	16.3	(38.4)	9.2	(26.1)	3.8	(13.8)	7.6	(22.3)
If PHQ > 10 ^α^	4.5	(5.4)	3.4	(3.9)	2.3	(2.3)	3.0	(3.4)
**PHQ Categories**								
Minimal	64.3	(49.8)	73.4	(39.9)	84.1	(26.2)	77.4	(35.1)
Mild	19.4	(41.1)	17.4	(34.2)	12.1	(23.4)	15.0	(29.9)
Moderate	9.8	(30.9)	5.6	(20.8)	2.6	(11.5)	4.8	(17.9)
Moderate-Severe and Severe	4.7	(22.0)	2.6	(14.4)	0.9	(6.8)	2.8	(13.9)
**Panel B**								
**Socio-demographic**								
Age (years)	43.5	(18.2)	48.4	(16.8)	48.6	(10.9)	47.6	(14.1)
**Age Categories**								
20–34 years	37.4	(50.3)	29.3	(41.1)	21.2	(29.3)	26.5	(37.0)
35–49 years	27.1	(46.2)	25.2	(39.2)	30.6	(33.0)	28.4	(37.8)
50–64 years	20.7	(42.1)	21.1	(36.8)	32.1	(33.4)	26.8	(37.2)
65+ years	14.9	(37.0)	24.4	(38.8)	16.1	(26.3)	18.4	(32.5)
**Female**	54.2	(51.8)	52.1	(45.1)	48.6	(35.8)	50.7	(41.9)
**Married**	45.3	(51.8)	58.1	(44.6)	73.0	(31.8)	63.7	(40.4)
**Educational attainment**								
Less than high school	37.2	(50.3)	21.3	(37.0)	6.0	(17.0)	16.1	(30.8)
High school graduate/GED	26.9	(46.1)	29.2	(41.0)	18.3	(27.7)	23.1	(35.3)
Some college or AA degree	28.2	(46.8)	34.5	(42.9)	32.0	(33.4)	32.1	(39.2)
College graduate or above	7.7	(27.8)	15.1	(32.3)	43.7	(35.5)	28.8	(38.0)
**Race/Ethnicity**								
White Non-Hispanic	53.3	(51.9)	68.9	(41.8)	85.6	(25.2)	74.9	(36.4)
Black Non-Hispanic	18.7	(40.6)	13.7	(31.1)	7.7	(19.1)	11.4	(26.7)
Hispanics	28.0	(46.7)	17.4	(34.2)	6.7	(17.9)	13.7	(28.8)
**Comorbidity**	1.1	(1.6)	1.1	(1.3)	0.9	(0.8)	98.2	(11.1)
**Has any health insurance coverage**	61.8	(50.5)	76.0	(38.6)	93.1	(18.2)	82.5	(31.9)
**Foreign Language-English**	88.1	(33.7)	94.9	(19.8)	99.4	(5.5)	96.0	(16.3)
**No rigorous physical activities**	58.4	(51.3)	50.4	(45.1)	34.5	(34.1)	43.5	(41.6)
**Smoking status**								
Never	45.9	(51.8)	50.3	(45.1)	57.7	(35.4)	53.4	(41.8)
Former	18.7	(40.6)	26.5	(39.9)	27.4	(32.0)	25.6	(36.6)
Current	35.3	(49.7)	23.2	(38.1)	14.8	(25.5)	21.0	(34.2)
**Drinking status**								
Never	15.8	(37.9)	13.2	(30.6)	6.7	(17.9)	10.2	(25.4)
Former	13.6	(35.6)	12.8	(30.1)	10.2	(21.7)	11.5	(26.8)
Current	70.6	(47.4)	74.0	(39.6)	83.1	(26.8)	78.2	(34.6)
**Fair/poor health**	33.6	(49.1)	22.0	(37.4)	8.7	(20.2)	17.1	(31.6)
**Live alone**	16.7	(38.8)	15.5	(32.7)	12.7	(23.8)	14.2	(29.3)

Note: Depression symptoms categories calculated using the Patient Health Questionnaire–9: none (0–4), mild (5–9), moderate (10–14), moderately severe (15–19), and severe (20). We combined moderately severe and severe as one group for this table. The Percentages are weighted to the population of noninstitutionalized U.S. adults aged 20 years or older. ^α^ We created a dummy variable if the score was equal to or higher than 10.

**Table 2 ijerph-19-06227-t002:** Weighted Negative Binomial Regression Estimates in U.S. Adults Aged 20 and above, NHANES 2005–2016.

	Model 1 (N = 23,949)		Model 2 (N = 23,949)	
**PIR Categories (Ref: if PIR > 2.83)**	IRR	[95% CI]	IRR	[95% CI]
Low-PIR (1.17–2.82)	1.55 ***	[1.46–1.65]	N/A	N/A
Medium PIR (2.83–5.00)	1.30 ***	[1.23–1.37]	N/A	N/A
Female	1.29 ***	[1.24–1.34]	N/A	N/A
**PIR Categories Interact By Sex** **(Ref: if High-PIR Women)**				
Low-PIR Women	N/A	N/A	1.54 ***	[1.44–1.65]
Medium-PIR Women	N/A	N/A	1.28 ***	[1.20–1.36]
Low-PIR men	N/A	N/A	1.19 ***	[1.10–1.29]
Medium-PIR men	N/A	N/A	1.01	[0.95–1.08]
High-PIR men	N/A	N/A	0.77 ***	[0.72–0.81]
**Age categories (Ref. 20–34 year)**				
35–49 yrs	1.07 *	[1.01–1.13]	1.07 *	[1.01–1.13]
50–64 yrs	0.96	[0.89–1.03]	0.96	[0.89–1.03]
65+ yrs	0.64 ***	[0.59–0.70]	0.64 ***	[0.59–0.70]
Married	0.79 ***	[0.75–0.84]	0.79 ***	[0.75–0.84]
**Education (Ref. Less than high school)**				
High school graduate/GED or equivalent	0.93 *	[0.87–0.99]	0.93 *	[0.87–0.99]
Some college or AA degree	0.89 **	[0.83–0.96]	0.89 **	[0.83–0.96]
College graduate or above	0.73 ***	[0.68–0.80]	0.73 ***	[0.68–0.80]
**Race/Ethnicity (Ref. NHW)**				
Black Non-Hispanic	0.92 **	[0.87–0.98]	0.92 **	[0.87–0.98]
Hispanics	0.96	[0.90–1.01]	0.96	[0.90–1.01]
**Comorbidity**	1.21 ***	[1.19–1.23]	1.21 ***	[1.19–1.23]
**Covered by health insurance**	0.95 *	[0.90–1.00]	0.95 *	[0.90–1.00]
**Language of Family Interview, EN.**	1.14 **	[1.04–1.24]	1.14 **	[1.04–1.24]
**Live alone**	0.99	[0.93–1.05]	0.99	[0.93–1.05]
**Constant**	2.32 ***	[2.06–2.61]	3.01 ***	[2.66–3.39]

* *p* < 0.05, ** *p* < 0.01, *** *p* < 0.001, IRR = Incidence-Rate Ratios, N/A = not applicable.

**Table 3 ijerph-19-06227-t003:** Weighted Negative Binomial Regression Estimates in Men and Women U.S. Adults Aged 20 and above, NHANES 2005–2016.

	Men (N = 12,064)	Women (N = 12,055)
	IRR [95% CI]	IRR [95% CI]
**PIR Categories (Ref: if PIR > 2.83)**		
Low-PIR (1.17–2.82)	1.61 ***	1.50 ***
	[1.48–1.75]	[1.39–1.63]
Medium-PIR (2.83–5.00)	1.34 ***	1.26 ***
	[1.24–1.45]	[1.18–1.35]
**Age categories (Ref. 20–34 yrs)**		
35–49 yrs	1.07	1.07
	[0.99–1.15]	[0.99–1.15]
50–64 yrs	1.01	0.90 *
	[0.91–1.13]	[0.83–0.98]
65+ yrs	0.65 ***	0.64 ***
	[0.58–0.74]	[0.58–0.70]
Married	0.78 ***	0.80 ***
	[0.71–0.85]	[0.75–0.86]
**Education (Ref. Less than high school)**		
High school graduate/GED or equivalent	1.01	0.86 **
	[0.92–1.11]	[0.78–0.95]
Some college or AA degree	0.98	0.82 ***
	[0.89–1.07]	[0.75–0.89]
College graduate or above	0.77 ***	0.70 ***
	[0.69–0.86]	[0.63–0.78]
**Race/Ethnicity (Ref. NHW)**		
Black Non-Hispanic	0.86 ***	0.98
	[0.80–0.93]	[0.92–1.05]
Hispanics	0.92 *	1.00
	[0.85–0.99]	[0.92–1.07]
**Comorbidity**	1.22 ***	1.21 ***
	[1.19–1.25]	[1.19–1.24]
Covered by health insurance	0.93	0.95
	[0.86–1.01]	[0.88–1.03]
Language of Family Interview, EN.	1.17 *	1.09
	[1.04–1.33]	[0.98–1.21]
Live alone	1.03	0.96
	[0.94–1.13]	[0.87–1.05]
Constant	2.09 ***	3.33 ***
	[1.77–2.47]	[2.84–3.90]

* *p* < 0.05, ** *p* < 0.01, *** *p* < 0.001. IRR = Incidence-Rate Ratios.

## Data Availability

Data available at: https://wwwn.cdc.gov/nchs/nhanes/.

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
