# Peer review of "How Income and Income Inequality Drive Depressive Symptoms in U.S. Adults, Does Sex Matter: 2005–2016"

_ijerph, 2022, doi:10.3390/ijerph19106227_

Round 1

Reviewer 1 Report

This is a sound paper. Scientifically relevant and methodologically robust, I have no particular observations on those merits. However, I found the scientific relevance and, in particular, the positioning of the paper within the debate sort of average. Why, in fact, should we be surprised by the fact that PIR show an association with a greater incidence of depression? Who is denying that in the US? What is the void or at least the niche in which this paper fits? This should be highlighted in the introduction, discussion and conclusions. Otherwise, you mostly get an exercise in modeling variables.

Author Response

Dear Reviewer, we thank you for your most valuable comments and appreciate having had this wonderful opportunity to learn from you. We hope that our responses have addressed your comments effectively. In the attached file we detailed in this letter how we revised the article.

This is a sound paper. Scientifically relevant and methodologically robust, I have no particular observations on those merits. However, I found the scientific relevance and, in particular, the positioning of the paper within the debate sort of average. Why, in fact, should we be surprised by the fact that PIR show an association with a greater incidence of depression? Who is denying that in the US? What is the void or at least the niche in which this paper fits? This should be highlighted in the introduction, discussion, and conclusions. Otherwise, you mostly get an exercise in modeling variables.

Response #1:

Thank you for the comment. We added a few lines to the Introduction to address this comment. We have copied them here:

A few studies have examined the association between income and depression; however, fewer studies have stratified analyses by sex. Among these studies, only three were based in the US, of which one was only among adolescents42 and the other two used less than 3 years of data.43, 44 This is the first study to examine the relationship between income inequality and depression in a wide range of NHANES data (2005-2016) and by sex.

Also following comments and questions raised by other reviewers and academic editors we modified the Introduction, Method section, and Discussion to improve the paper.

Reviewer 2 Report

Dear Authors; I find this an interesting study on investigation of the relationship of PIR and depression index among American population and its overall standing is solid. However, its  current presentation status is 2 degree below journal standards and needs some "serios work" to arrive to journal standards. Regards.

P.S.

[1] Writing

1-1 Author Affiliation: add city, state, country

1-2 References: make them MDPI format. Example: years for the papers are in bold font.

1-3 List of Abbreviations: This is international journal and readers across globe are reading it and deserve minimum effort to figure out your American abbreviations. List Abbreviations used in the work right before reference section. Example:

Abbreviations

PIR: Property Income Ratio; etc.

1-4 Line 70: Missing "niche" for this study : What is missing in the current literature that this work fills in ? What is advantage of considering PIR as predictor in which other publications didn't ? Need to add a paragraph on it.

1-5 Line 149: What is GC abbreviation for ? add it.

1-6 Discussion: Its current status is "messy". Divide it in three parts: 4.Discussion, 4.1. This Work 4.1.1. title, 4.1.2. title, 4.1.3. title, 4.2. Limitations, 4.3. Future Work

1-7. Author contributions statement: Format in MDPI template.

[2] Statistical:

2-1. Missing Software citation: Seems to me you used STATA. In case add it usage in 156 and add the following citation to your reference section:

[Citation] StataCorp. 2021. Stata Statistical Software: Release 17. College Station, TX: StataCorp LLC. 

2-2. Parsimonious regression models: The default modelling choice for your data is "Poisson Regression Model". It is discarded in favour of your used "NB Regression Model" when there is overdispersion/underdispersion in the model. . Add a paragraph there in line 124, mention using  "Poisson Regression Model" and report the overdispersion/underdispersion statistics in that model. Then move on with current reported analysis. 

Author Response

Dear Reviewer, we thank you for your most valuable comments and appreciate having had this wonderful opportunity to learn from you. We hope that our responses have addressed your comments effectively. In the attached file we detailed in this letter how we revised the article.

Dear Authors; I find this an interesting study on investigation of the relationship of PIR and depression index among American population and its overall standing is solid. However, its current presentation status is 2 degree below journal standards and needs some "serious work" to arrive to journal standards. Regards.

Response #1a:

Thank you for the detail-oriented comments, we have addressed all of these comments using track changes version.  

P.S.

[1] Writing

1-1 Author Affiliation: add city, state, country

1-2 References: make them MDPI format. Example: years for the papers are in bold font.

1-3 List of Abbreviations: This is an international journal and readers across the globe are reading it and deserve minimum effort to figure out your American abbreviations. List Abbreviations used in the work right before the Reference section. Example:

Abbreviations

PIR: Property Income Ratio; etc.

Response #1b: We added the list of abbreviations.

:

1-4 Line 70: Missing "niche" for this study: What is missing in the current literature that this work fills in? What is the advantage of considering PIR as predictor in which other publications didn't? Need to add a paragraph on it.

1-5 Line 149: What is GC abbreviation for? add it.

Response #1c: We added a few lines to explain the PIR, we have copied here:

PIR as a socioeconomic status indicator compared to income and explains more about the family income by including the family income, family size, and the poverty threshold, providing more reliable socioeconomic status than income.

We also defined the Gini Coefficient.

1-6 Discussion: Its current status is "messy". Divide it in three parts: 4.Discussion, 4.1. This Work 4.1.1. title, 4.1.2. title, 4.1.3. title, 4.2. Limitations, 4.3. Future Work

Response #1d: We reformatted the discussion to address this comment.

1-7. Author contributions statement: Format in MDPI template.

Response #1e: We reformatted the statement following the journal guidelines.

[2] Statistical:

2-1. Missing Software citation: Seems to me you used STATA. In case add it usage in 156 and add the following citation to your reference section:

[Citation] StataCorp. 2021. Stata Statistical Software: Release 17. College Station, TX: StataCorp LLC. 

Response #1f: Thank you for noticing this. We added the information for the Software.

2-2. Parsimonious regression models: The default modelling choice for your data is "Poisson Regression Model". It is discarded in favor of your used "NB Regression Model" when there is overdispersion/underdispersion in the model. Add a paragraph there in line 124, mention using “Poisson Regression Model" and report the overdispersion/underdispersion statistics in that model. Then move on with current reported analysis. 

Response #2: Thank you. We added the following paragraph to the paper, please see page. 3.

To find the best fit model, first, we ran a Poisson regression model, even though we believe that the Poisson distribution is incorrect. The considerable value of goodness of fit chi-square (24103) and p<0.01 indicates that the Poisson model is inappropriate.53 After running NBRG, by looking at the likelihood ratio test (a test of the overdispersion parameter alpha). Alpha was significantly different from zero (P<0.01) and thus reinforces one last time that the Poisson distribution is not appropriate, and the Negative binomial regression is more appropriate in cases of overdispersion.

Reviewer 3 Report

  Dear authors

I would like to congratulate you on your article and express my admiration for your efforts and endeavors in relation to the manuscript : How Income and Income Inequality Drives Depressive Symp- 2
toms in U.S. Adults, Does Sex Matter: 2005-2016  

I appreciate the structure of the study as well as your research and its processing.
I also appreciate results and discussion   of the paper.    I suggest you to add studies of several authors, who´s work will expand (by reference) your study with other assumptions and observations.    For example:    I recommend publishing the text after your choice   Bolea, Ștefan . (2020). The Courage To Be Anxious. Paul Tillich’s Existential Interpretation of Anxiety. Journal of Education Culture and Society, 6(1), 20–25. https://doi.org/10.15503/jecs20151.20.25    Kobylarek, A., Błaszczyński, K., Ślósarz, L., Madej, M., Carmo, A., Hlad, Ľ., Králik, R., Akimjak, A., Judák, V., Maturkanič, P., Biryukova, Y., Tokárová, B., Martin, J. G., & Petrikovičová, L. (2022). The Quality of Life among University of the Third Age Students in Poland, Ukraine and Belarus. Sustainability, 14(4), 2049. https://doi.org/10.3390/su14042049    Petrovič, F., Murgaš, F., & Králik, R. (2021). Happiness in Czechia during the COVID-19 Pandemic. Sustainability, 13(19), 10826. https://doi.org/10.3390/su131910826      

Author Response

Dear Reviewer, we thank you for your most valuable comments and appreciate having had this wonderful opportunity to learn from you. We hope that our responses have addressed your comments effectively. In the attached file we detailed in this letter how we revised the article.

I would like to congratulate you on your article and express my admiration for your efforts and endeavors in relation to the manuscript: How Income and Income Inequality Drives Depressive Symp- 2
toms in U.S. Adults, Does Sex Matter: 2005-2016  

I appreciate the structure of the study as well as your research and its processing.
I also appreciate results and discussion of the paper. I suggest you to add studies of several authors, who´s work will expand (by reference) your study with other assumptions and observations. For example:  I recommend publishing the text after your choice  

Bolea, Ștefan . (2020). The Courage To Be Anxious. Paul Tillich’s Existential Interpretation of Anxiety. Journal of Education Culture and Society6(1), 20–25. https://doi.org/10.15503/jecs20151.20.25   

Kobylarek, A., Błaszczyński, K., Ślósarz, L., Madej, M., Carmo, A., Hlad, Ľ., Králik, R., Akimjak, A., Judák, V., Maturkanič, P., Biryukova, Y., Tokárová, B., Martin, J. G., & Petrikovičová, L. (2022). The Quality of Life among University of the Third Age Students in Poland, Ukraine and Belarus. Sustainability14(4), 2049. https://doi.org/10.3390/su14042049   

Petrovič, F., Murgaš, F., & Králik, R. (2021). Happiness in Czechia during the COVID-19 Pandemic. Sustainability13(19), 10826. https://doi.org/10.3390/su131910826      

Response #1:  Thank you very much for the encouraging words and for sharing these articles we have added a few sentences and cited these works in a few places in the MS. For example, we add a paragraph to the introduction:

Financial loss from retirement can lead to lower incomes for the elderly population and the loss of restriction of social interactions might worsen depressive symptoms.32 Lower-income has been commonly associated with worse physical health.33-35 Robust studies have highlighted that chronic conditions are associated with depressive disorders.36, 37   

We also use the suggested citation in discussion and modified the discussions to address the reviewer and other reviewers’ comments.

Round 2

Reviewer 2 Report

Dear Authors, most of my concerns were addressed satisfactorily. Regards.